# Evaluation of Nutritional Status in an Acute Geriatric Unit: Retrospective Study and Analysis of Frailty Syndrome

**DOI:** 10.3390/medicines10030022

**Published:** 2023-03-08

**Authors:** Abrar-Ahmad Zulfiqar, Ibrahima Amadou Dembele, Emmanuel Andres

**Affiliations:** Département de Médecine Interne, CHU Strasbourg, Clinique Médicale B, 67000 Strasbourg, France

**Keywords:** frailty, malnutrition, elderly, Fried scale, SEGA score, Rockwood scale

## Abstract

Introduction: The aim of our study is to evaluate the nutritional status of patients in an acute geriatric unit. Methods: Patients included in the study were hospitalized in an acute geriatric unit over a period of 6 months. The nutritional status of each patient was evaluated with anthropometric measurements (the BMI and MNA scales), and biological measurements (albumin). Frailty was evaluated using three scales: the Fried scale, the CFS and the modified SEGA scale. Results: A total of 359 patients were included, comprising 251 women (70%) with an average age of 85.28 years. The study showed that 102 elderly subjects were considered undernourished according to the BMI scale, 52 subjects were undernourished according to the MNA scale, and 50 subjects were undernourished according to their albumin levels. The relationships between undernutrition and frailty syndrome studied in our work show that elderly subjects who are undernourished according to the BMI and MNA scales are significantly frail according to Fried and Rockwood, whereas those who are undernourished according to their albumin levels are significantly frail according to Fried and the modified SEGA scale. Conclusion: The relationship between undernutrition and the frailty syndrome is close, and their joint screening is necessary, whether on an outpatient or in-hospital basis, in order to prevent negative events related to comorbidities and geriatric syndromes.

## 1. Introduction

Malnutrition is defined as an imbalance between nutritional intake and the body’s needs. It therefore has an exogenous component, linked to an insufficient intake, and an endogenous component, linked to an increase in needs. Its frequency increases with age, making this pathology one of the most frequently encountered in geriatrics. In the elderly, we observe a sensory alteration that leads to a decrease in taste and smell, along with polypharmacy and diets linked to pathologies, accompanied by disordered appetite regulation. This leads to a decrease in intake, responsible for chronic anorexia that increases the basic metabolism, which may already be increased by the presence of a pathology or an inflammatory state, thus leading to a further increase in needs [1,2,3,4]. Several studies have shown that this geriatric pathology concerns 20 to 50% of hospitalized patients [5,6,7]. While we know that nutritional status is a major factor in morbidity and mortality, research has also shown undernutrition to be a predictor of both short- and long-term mortality [8,9,10].

We wanted to focus on another preponderant factor of morbidity and mortality: frailty syndrome. This term was coined about thirty years ago, its definition remains controversial, and no consensus has been found around it to date. Fried defines it as “a vulnerability linked to advancing age, due to an alteration of the homeostatic reserves of the organism which becomes incapable of overcoming any stress” [11]. The difficulty in finding a consensual definition lies in the fact that it is a dynamic, evolving and multidimensional concept encompassing physical, physiological, biological, social and environmental factors [12,13,14,15]. All authors agree that it is a major morbidity and mortality factor. Some of these factors are reversible, making their detection and prevention a major public health issue. According to the Haute Autorité de Santé (HAS) [16], two models of frailty can be distinguished in the literature: the model based on Fried’s so-called functional frailty, which takes into account the concept of sarcopenia [11]; and the model based on multidimensional frailty, which takes into account social, environmental and cognitive factors [17].

Early detection of these two potentially reversible factors is a pressing public health issue, as multidisciplinary management can help reduce certain avoidable adverse effects of aging, and therefore expenses [18]. The SAFES study (Sujets Agés Fragiles Evaluation et Suivi) found that pre-frail and fragile individuals tend to have longer hospital stays and higher rates of re-hospitalization, as well as an increased risk of hospitalization and death [10]. Another study showed that preventing frailty could decrease mortality by 3–5% [19]. The purpose of this study was to determine the prevalence of these two major geriatric syndromes in our service, and to investigate the connection between them.

## 2. Materials and Methods

### 2.1. Type of Study

This is a descriptive retrospective study on the consultation of medical records in patients over the age of 65 and hospitalized in an acute geriatrics unit. This work was carried out in the Department of Geriatric Medicine and Internal Medicine 1 (UMG 1), Maison Blanche Hospital, Center Hospitalier Universitaire de Reims, from 24 November 2014 to 24 May 2015. Patients labeled under palliative care when entering the unit were excluded.

### 2.2. Data Collection

For each patient hospitalized in the department during this period, we collected: sex and age, reason for hospitalization, medical and surgical history, and Charlson score. On the social level, we collected: presence of an entourage; way of life (origin home, nursing home, other (home of children or grandchildren)); and incidences of hospitalization over the last two years.

On the biometric level, we collected: brachial circumference (normal value > 21 cm); calf circumference (normal value >31 cm); height (using the heel/knee height when measurement in a standing position was impossible); and weight. From these values, we were able to calculate the Body Mass Index (BMI) of each subject from the formula: weight (in kg)/(height × height (in m). Concerning its interpretation, we used the normal HAS BMI values if > or = 24 kg/m^2^. For a multifactorial approach we also carried out the Mini Nutritional Assessment (MNA), which allowed us to define whether the patients were: not malnourished (score > 24); at risk of malnutrition (score between 17 and 24); or malnourished (score < 17).

Data from the Comprehensive Geriatric Assessment (CGA) were also collected: the monopod support test, which was defined as positive if the monopod support was greater than or equal to 5 s. Dependence was assessed using the Katz and Lawton scales: Activity Daily Living (ADL) and Instrumental Activity Daily Living (IADL). Memory disorders were estimated by the Mini Mental State (MMS) score. The study of thymia was carried out with the help of the mini-Geriatric Depression Scale (GDS), with a score greater than or equal to 1 indicating a high probability of depression.

From a biological standpoint, we noted the values of the entry assessment of each patient upon admission into the department, carried out on Day 1 by the nurses: albumin level (hypoalbuminemia was defined by an albuminemia strictly lower than 35 g/L); creatinine clearance (renal failure was defined as creatinine clearance < 60 mL/min/1.73 m^2^); hemoglobin (anemia was defined as hemoglobin < 12 g/dL); lymphocytes (lymphopenia corresponds to a rate of lymphocytes <1 G/L); thyroid-stimulating hormone (TSH) (hypothyroidism was defined by a TSH > 4 microU/mL, hyperthyroidism by a TSH < 0.4 microU/mL); C Related Protein (CRP) (the biological inflammatory syndrome corresponds at a CRP >5 mg/L); vitamin B12 (vitamin B12 deficiency with a level lower than 191 pg/mL and hypervitaminosis B12 with a level >663 pg/mL); and vitamin D (a deficiency in which corresponds to a level less than 30 ng/mL).

To define frailty, we used the Fried scale [11], the modified SEGA scale (mSEGA) [20], and the CFS (Clinical Frailty Scale) [21]. On the Fried scale, individuals with a score of 0 were considered non-frail, those with a score between 1 and 2 were considered pre-frail, and those with a score of 3 or higher were considered fragile. On the mSEGA part A scale, individuals with a score of less than or equal to 8 were considered not very frail, those with a score between 8 and 11 were considered frail, and those with a score greater than 11 were considered very frail. On the Rockwood scale (CFS), for scores of 5 or more, the elderly patient was considered by CFS to be “frail”. The scales used in screening for frailty syndrome are all validated, in particular by expert committees such as the Haute Autorité de Santé in France. The Fried scale remains the reference on the phenotypic side, the Rockwood scale is the reference on the multidimensional side, and the SEGA scale remains a multidimensional scale validated in elderly subjects and used mainly by French-speaking practitioners.

### 2.3. Statistical Analyses

For the research method, we conducted a single-factor analysis of the various factors that may impact the scale of frailty using the Kaplan–Meier method. The multifactorial analysis was conducted using the Cox model with an ascending “step by step” selection of candidate variables. The statistical tests were performed using SAS V9.1 software (SAS Institute, Inc., Cary, NC, USA) with a significance level of 0.05.

### 2.4. Ethical Considerations

This study was conducted according to the guidelines set by the Declaration of Helsinki and was registered with the CNIL (National Commission for Computing and Liberties) (CNIL registration no.: 1880518 v 0).

## 3. Results

A total of 359 patients were included in the study, comprising 251 women (70%) with an average age of 85.28 years. Regarding existing pathologies, cardiological conditions were the most common, affecting 308 subjects (86%), followed by mental health conditions in 121 patients (34%), and type 2 diabetes in 78 patients (22%). The majority of patients came from home (259 patients or 72%), while 68 came from a retirement home (19%). An entourage (a support system of family or caregivers) was present for 298 subjects (83%). Out of the total number of patients, 197 (55%) had been hospitalized within the past two years. We also observed oral mycotic pathology in 44 patients (12%), with 65 subjects (18%) showing normal results on a monopod weight bearing test. Biologically, we found that 190 subjects (53%) had hypoalbuminemia, 155 patients (43%) had renal failure, 25 patients (7%) had hypothyroidism, 176 patients (49%) had anemia, 77 subjects (21%) had lymphopenia, 241 patients (67%) had a biological inflammatory syndrome, 250 patients (70%) had vitamin D deficiency, 25 patients (7%) had vitamin B12 deficiency, and 88 patients (25%) had hypervitaminemia B12. Pressure ulcers were detected in 52 elderly patients (14%), with 3 patients (1%) having type 1, 26 patients (7%) having type 2, 19 patients (5%) having type 3, and 4 subjects (1%) having type 4. At the end of their hospitalization, 127 subjects returned home (35%), 65 went to a retirement home (18%), and 126 subjects were transferred to a rehabilitation center. One patient died after the hospitalization of 33 patients (9%), another patient died after the hospitalization of 60 patients (17%), and 89 patients (25%) were hospitalized again. The descriptive analyses are provided in Table 1. A total of 102 elderly subjects were considered malnourished according to the BMI scale, 52 subjects were considered malnourished according to the MNA scale, and 50 subjects were considered malnourished based on their albumin levels.

The relationship between undernutrition and frailty syndrome studied in this research showed that elderly subjects who were undernourished according to the BMI and MNA scales were significantly frail according to the Fried the Rockwood scores, while those who were undernourished according to their albumin levels were significantly fragile according to the Fried score and the modified SEGA scale (see Table 2). Table 3 presents the results of the relationship between undernutrition and the status of vitamins D and B12. These results show that there is no statistically significant link between malnutrition based on BMI, MNA, and albumin level, and vitamin deficiency, aside from that between the albumin criterion (*p* = 0.0108) and hypervitaminemia B12. In terms of geriatric criteria, there is a statistically significant relationship between undernutrition based on the three measurement types and loss of autonomy based on ADL and IADL, but there is no significant relationship between undernutrition and the Charlson comorbidity score. There is a significant relationship between undernutrition based on the MNA scale and altered thymic state based on the miniGDS scale, as well as between this measurement of undernutrition and cognitive level as measured by the Mini Mental State (MMS), as confirmed in Table 4. There is no statistically significant relationship linking undernutrition and the three measurement types, the monopod weight-bearing test, or the “Hospitalization in the last two years” criterion, aside from the albumin measurement (see Table 5).

## 4. Discussion

Our research focused on hospitalized elderly subjects, focusing on measurements of malnutrition and the evaluation of frailty syndrome, which has been rarely studied in the scientific literature. This research represents a novelty in the sense that several geriatric scales (malnutrition, frailty syndrome, autonomy/dependence, thymia, cognitive assessment and biological markers) have been used together in hospitalized elderly subjects, which was not achieved to this extent in previous scientific work. Our results showed that there is a statistically strong relationship between undernutrition based on the BMI and MNA scales and frailty as measured by both the Fried phenotypic scale and the Rockwood multidimensional scale, while undernutrition based on albumin level is significantly related to frailty according to the Fried modified SEGA scales. The relationship between undernutrition and frailty syndrome has been demonstrated in previous scientific studies, as indicated by a systematic review of the literature [22]. Our work highlights the importance of measuring and evaluating undernutrition, as well as frailty syndrome, as they can predict negative events and an increase in morbidity and mortality. For example, a Japanese study in the context of managing heart failure found that malnutrition and underweight were significant predictors of adverse outcomes in elderly patients with heart failure and surgically untreated moderate-to-severe functional mitral regurgitation [23]. Frailty syndrome is also characterized by weight loss, a decline in functional abilities and physical strength, and geriatric consequences such as falls, protein-energy malnutrition, and memory problems. Sarcopenia is a generalized and progressive loss of skeletal muscle mass, strength, and function that can be caused by the primary effects of aging and secondary effects of other factors, such as diseases, malnutrition, and inactivity [24]. The goal of this research is to raise awareness about the need to detect malnutrition and frailty in hospitalized elderly subjects, in order to treat or prevent the worsening of geriatric comorbidities and reduce the duration of hospitalizations, which can cause increased morbidity and mortality. Malnutrition is a significant burden for the elderly population and is becoming a priority for promoting healthy aging. Malnutrition plays a key role in the pathophysiology of frailty syndrome. In one study, malnutrition and the risk of malnutrition were associated with a nearly four-fold increase in the risk of frailty. Nutritional risk also increases the consequences of frailty, including the risk of hospitalization and loss of independence [25]. Nutritional evaluation is an important component of frailty screening because frail individuals are a primary target for nutritional intervention [26].

In a study by Jurschik involving 640 individuals, who were on average 81 years old and living at home, it was found that most frail individuals were at risk of malnutrition and those who were malnourished were considered frail, with a significant relationship between the five frailty criteria according to Fried and the MNA [27]. Another French study involving 267 autonomous patients without cognitive disorders or cancerous pathology, who were on average 81.5 years old, also found a significant relationship between the MNA and frailty [26]. A study by Dorner involving 133 elderly people hospitalized in internal medicine, who were on average 74 years old, found a relationship between frailty and the risk of malnutrition (*p* < 0.001) and between frailty and malnutrition (*p* < 0.001) [28]. The DENT study, involving 100 patients hospitalized in an acute geriatric unit, found a significant association between frailty according to Fried and the MNA (r −0.479, *p* < 0.001) [29]. Nevertheless, no scientific study has shown a superiority in the link with screening for frailty and the measurement of malnutrition by the MNA scale compared to other assessment measures, such as the BMI scale and albumin level. This deserves additional research.

### Limitations

One of the main limitations of our study is its retrospective and single-center design, even with a large number of staff. The retrospective nature of this study also prevents us from conducting predictive research on potential negative events over a long follow-up period. In addition, we did not examine the oral condition of all subjects, which is essential for understanding undernutrition and its relationship with the frailty syndrome. The frailty syndrome screening scales also have limitations in their use. Indeed, they can be time-consuming, as is the case with the SEGA scale, and especially with the Fried scale, which requires the use of a dynamometer. Regarding the CFS, or Rockwood scale, it requires in-depth geriatric knowledge and thus requires prior training; however, it is thus a dependent evaluator.

## 5. Conclusions

The relationship between undernutrition and frailty syndrome is close, and their joint screening is necessary, whether on an outpatient or in-hospital basis, in order to prevent negative events related to comorbidities and geriatric syndromes. Despite its limitations, our study highlights the need for further research with the aim of understanding the underlying physiopathology of frailty syndrome and its causes and consequences, including undernutrition.

## Figures and Tables

**Table 1 medicines-10-00022-t001:** Characteristics of the general population.

	Medium	Median	Minimum	Maximum	[Q1; Q3]	Standard Deviation	Workforce (N = 359)	Number Missing Values
Age (y)	85.28	86	65	102	[81; 90]	7	359	-
Weight (kg)	63.61	63	31	108	[52.8; 73.2]	14.7	358	1
Height (m)	1.575	1.57	1.34	1.83	[1.52; 1.63]	0.1	358	1
Knee Heel Height (cm)	47,33	47	35	60	[45; 50]	4.2	354	5
BMI (kg/m^2^)	25.66	25.16	11.73	55.14	[21.61; 29.43]	5.9	358	1
Bicep Circumference (cm)	24.53	24	12.5	41	[22; 27]	4.4	358	1
Calf circumference (cm)	29.98	30	16	47	[27; 33]	4.8	358	1
Thigh Circumference (cm)	39.32	39	22.5	66	[35; 44]	6.5	358	1
MMSE (/30)	16.63	17	0	30	[12; 23]	7.3	307	52
MNA (/30)	21.4	22.5	0	29	[18.5; 25]	4.7	312	47
ADL (/6)	3.5	3.5	0	6	[0; 6]	2.4	359	-
IADL (/8)	4.401	1	0	8	[0; 4]	2.7	359	-
Albumin (g/L)	35.16	35	19	64	[32; 38]	5.3	356	3
Creatinin	104.5	80	23	583	[63; 118]	78.3	359	-
Clearance (mL/min/1.73 m^2^)	60.33	65	6	172	[42; 80]	25.1	359	-
Glycemia (g/L)	6.21	5.7	0.69	16.4	[5; 6.8]	2.1	358	1
TSH (microU/mL)	2.227	1.59	0.01	60	[1.02; 2.37]	3.8	357	2
Hemoglobin (g/dL)	11.84	12.0	13	17.1	[10.8; 13.2]	19.2	359	-
Lymphocytes (G/L)	1.743	1.4	0.3	55.5	[1; 1.9]	3.1	359	-
CRP (mg/L)	59.52	27.3	0.3	525	[5.6; 81]	79.6	359	-
Vitamin D (ng/mL)	22.55	20	5	95	[11; 30]	13.8	340	19
Vitamin B12 (pg/mL)	569.3	418	34	2000	[297; 658]	451.7	354	5
miniGDS (/4)	2.5	3	0	4	[2; 4]	1.3	316	43
mSEGA (/26)	14.86	15	2	26	[11; 18]	4.9	359	-
Fried (/5)	3.56	4	0	6	[3; 5]	1.4	359	-
Rockwood (/7)	5.67	6	1	7	[5; 7]	1.4	359	-
Charlson	7.262	7	1	16	[6; 8]	2.0	359	-

BMI—Body Mass Index; MMSE—Mini Mental State Examination; MNA—Mini Nutritional Assessment; ADL—Activity Daily Living; IADL—Instrumental Activity Daily Living; TSH—Thyroid Stimulating Hormone; CRP—C-reactive protein; miniGDS—mini Geriatric Depression Scale; mSEGA—modified Short Emergency Geriatric Assessment.

**Table 2 medicines-10-00022-t002:** Relationship between undernutrition and frailty.

		Workforce		Medium		Standard Deviation		Median		[Q1; Q3]		Chi-Square Test
		Malnourished	Non Malnourished	Malnourished	Non Malnourished	Malnourished	Non malnourished	Malnourished	Non Malnourished	Malnourished	Non Malnourished	*p*-Value
BMI	*Fried*											0.0026
	Frail	88 (33%)	182 (67%)	4.28	4.23	0.84	0.83	5	4	[3; 5]	[4; 5]	
	Non frail	14 (16%)	74 (84%)	1.57	1.45	0.65	0.71	2	2	[1; 2]	[1; 2]	
	*SEGA*											0.0665
	Frail	96 (30%)	224 (70%)	16.83	15.53	3.75	4.00	17	15	[14; 20]	[13; 18]	
	Non frail	6 (16%)	32 (84%)	5.50	6.22	1.52	1.60	5	7	[4; 6]	[5; 7]	
	*Rockwood*											0.0124
	Frail	91 (31%)	199 (69%)	6.38	6.16	0.73	0.78	7	6	[6; 7]	[6; 7]	
	Non frail	11 (16%)	57 (84%)	3.27	3.30	0.90	0.71	4	3	[2; 4]	[3; 4]	
MNA	*Fried*											0.0071
	Frail	46 (20%)	183 (80%)	4.30	4.15	0.84	0.84	5	4	[4; 5]	[3; 5]	
	Non frail	6 (7%)	77 (93%)	0.83	1.55	0.98	0.64	0	2	[0; 2]	[1; 2]	
	*SEGA*											0.1368
	Frail	49 (18%)	226 (82%)	17.94	15.14	3.80	3.69	18	15	[16; 20]	[12; 18]	
	Non frail	3 (8%)	34 (92%)	7.67	5.94	0.58	1.59	8	6	[7; 8]	[5; 7]	
	*Rockwood*											0.0092
	Frail	48 (20%)	198 (80%)	6.52	6.10	0.68	0.77	7	6	[6; 7]	[5; 7]	
	Non frail	4 (6%)	62 (94%)	3.25	3.29	0.96	0.73	3	3	[2; 4]	[3; 4]	
Albumin	*Fried*											0.0012
	Frail	47 (17%)	223 (83%)	4.43	4.20	0.8	0.84	5	4	[4; 5]	[3; 5]	
	Non frail	3 (3%)	83 (97%)	0.67	1.48	0.58	0.69	1	2	[0; 1]	[1; 2]	
	*SEGA*											0.00119
	Frail	48 (15%)	269 (85%)	16.94	15.73	4.5	3.85	17	16	[13; 20]	[13; 19]	
	Non frail	2 (5%)	37 (95%)	8.00	6.05	0	1.58	8	6	[8; 8]	[5; 7]	
	*Rockwood*											0.0894
	Frail	47 (16%)	240 (84%)	6.47	6.18	0.72	0.78	7	6	[6; 7]	[6; 7]	
	Non frail	3 (4%)	66 (96%)	3.33	3.30	0.58	0.74	3	3	[3; 4]	[3; 4]	

BMI—Body Mass Index; MNA—Mini Nutritional Assessment; SEGA—Short Emergency Geriatric Assessment.

**Table 3 medicines-10-00022-t003:** Relationship between undernutrition and vitamin B12/vitamin D.

		Workforce		Chi-Square Test
		Malnourished	Non Malnourished	*p*-Value
BMI				
	*Vitamin D deficiency*			0.4436
	Yes	68 (27%)	182 (73%)	
	No	28 (31%)	61 (69%)	
	*B12 hypervitaminosis*			0.1944
	Yes	43 (32%)	90 (68%)	
	No	57 (26%)	163 (74%)	
MNA				
	*Vitamin D deficiency*			0.0926
	Yes	32 (15%)	185 (85%)	
	No	18 (23%)	60 (77%)	
	*B12 hypervitaminosis*			0.0891
	Yes	25 (22%)	91 (78%)	
	No	27 (14%)	165 (86%)	
Albumin				
	*Vitamin D deficiency*			0.834
	Yes	34 (14%)	214 (86%)	
	No	13 (15%)	76 (85%)	
	*B12 hypervitaminosis*			0.0108
	Yes	26 (20%)	106 (80%)	
	No	22 (10%)	197 (90%)	

BMI—Body Mass Index; MNA—Mini Nutritional Assessment.

**Table 4 medicines-10-00022-t004:** Relationship between undernutrition and geriatric factors.

		Medium		Standard Deviation		Median		[Q1; Q3]		Wilcoxon Test
		Malnourished	Non Malnourished	Malnourished	Non Malnourished	Malnourished	Non Malnourished	Malnourished	Non Malnourished	*p*-Value
ADL										
	*BMI*	2.33	3.46	2.41	2.35	2.0	4.0	[0; 5]	[1; 6]	6.391 × 10^−5^
	*MNA*	2.02	3.65	2.17	2.30	1.0	4.0	[0; 3.5]	[2; 6]	1.324 × 10^−5^
	*Albumin*	2.30	3.29	2.39	2.40	1.0	3.5	[0; 4.5]	[0; 6]	0.0040
IADL										
	*BMI*	1.73	2.67	2.32	2.81	0	2	[0; 3]	[0; 5]	0.0024
	*MNA*	1.19	2.90	1.75	2.83	0	2	[0; 2]	[0; 5]	7.792 × 10^−5^
	*Albumin*	1.34	2.59	1.97	2.78	0	2	[0; 2]	[0; 5]	0.0020
miniGDS										
	*BMI*	2.49	2.50	1.34	1.33	3	3	[2; 4]	[2; 4]	0.9457
	*MNA*	1.78	2.65	1.28	1.32	2	3	[1; 3]	[2; 4]	6.355 × 10^−5^
	*Albumin*	2.24	2.53	1.28	1.33	2	3	[1; 4]	[2; 4]	0.1636
MMSE										
	*BMI*	15.85	16.92	8.06	7.08	17	18	[9; 23]	[12; 23]	0.3603
	*MNA*	11.53	18.24	7.29	6.51	11	19	[6; 17]	[14; 23]	3.860 × 10^−8^
	*Albumin*	14.80	16.89	7.22	7.33	14	18	[9; 20]	[12; 23]	0.0899
Charlson										
	*BMI*	7.23	7.28	2.26	1.96	7	7	[6; 8]	[6; 8]	0.642
	*MNA*	7.23	7.16	1.68	2.02	7	7	[6; 8]	[6; 8]	0.4871
	*Albumin*	7.32	7.27	1.75	2.08	7	7	[6; 8]	[6; 8]	0.6918

BMI—Body Mass Index; MMSE—Mini Mental State Examination; MNA—Mini Nutritional Assessment; ADL—Activity Daily Living; IADL—Instrumental Activity Daily Living; miniGDS—mini Geriatric Depression Scale; mSEGA—modified Short Emergency Geriatric Assessment.

**Table 5 medicines-10-00022-t005:** Relationship between undernutrition, cognitive disorders and normal monopod test.

		Workforce		Chi-Square Test
		Malnourished	Non Malnourished	*p*-Value
BMI				
	*Cognitive disorders*			0.9223
	Yes	65 (27%)	173 (73%)	
	No	27 (28%)	70 (72%)	
	*Normal monopod support test*			0.1698
	Yes	14 (22%)	51 (78%)	
	No	88 (30%)	205 (70%)	
	*Hospitalization in the last 2 years*			0.3282
	Yes	60 (31%)	136 (69%)	
	No	42 (26%)	120 (74%)	
MNA				
	*Cognitive disorders*			0.0018
	Yes	45 (22%)	159 (78%)	
	No	7 (7%)	88 (93%)	
	*Normal monopod support test*			0.2251
	Yes	7 (11%)	54 (89%)	
	No	45 (18%)	206 (82%)	
	*Hospitalization in the last 2 years*			0.2028
	Yes	33 (19%)	140 (81%)	
	No	19 (14%)	120 (86%)	
Albumin				
	*Cognitive disorders*			0.2928
	Yes	35 (15%)	202 (85%)	
	No	10 (10%)	86 (90%)	
	*Normal monopod support test*			0.1030
	Yes	5 (8%)	60 (92%)	
	No	45 (15%)	246 (85%)	
	*Hospitalization in the last 2 years*			0.0094
	Yes	36 (18%)	160 (82%)	
	No	14 (9%)	146 (91%)	

BMI—Body Mass Index; MNA—Mini Nutritional Assessment.

## Data Availability

The datasets used and/or analyzed during the current study are available from the corresponding author upon reasonable request.

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
