# Peer review of "Evaluation of Nutritional Status in an Acute Geriatric Unit: Retrospective Study and Analysis of Frailty Syndrome"

_medicines, 2023, doi:10.3390/medicines10030022_

Round 1
Reviewer 1 Report
The manuscript is well written, but I would like that the authors clarify the two points below:
1. Introduction – authors write, that elderly have the same nutritional needs as adults, but i.a. intake of elderly is decreased. But as we know energy requirements tend to decrease with age, so on a per-unit energy basis, nutrient needs increase, in other words, the nutrient density of older people's diets should be increased.
2. Discussion – why authors think, that MNA is the best tool? The existence of a relationship between frailty syndrome (FS) and MNA does not prove that it’s the best tool, especially since there is also a relationship between FS and other methods (BMI, albumin) that also can be used. The fact that in other studies there is also a relationship between FS and MNA does not prove this thesis either (best tool). Please explain this.
Author Response
RESPONSE TO REVIEWER 1 :
The manuscript is well written, but I would like that the authors clarify the two points below:
- Introduction – authors write, that elderly have the same nutritional needs as adults, but i.a. intake of elderly is decreased. But as we know energy requirements tend to decrease with age, so on a per-unit energy basis, nutrient needs increase, in other words, the nutrient density of older people's diets should be increased.
RESPONSE: I agree with you and we rewrite the sentence.
- Discussion – why authors think, that MNA is the best tool? The existence of a relationship between frailty syndrome (FS) and MNA does not prove that it’s the best tool, especially since there is also a relationship between FS and other methods(BMI, albumin) that also can be used. The fact that in other studies there is also a relationship between FS and MNA does not prove this thesis either (best tool). Please explain this.
RESPONSE : This is a hasty conclusion on our part regarding the MNA scale. This scale is very useful in screening for undernutrition and the link between undernutrition and the frailty syndrome has been proven. Nevertheless, no scientific study has shown a superiority in the link with screening for frailty and the measurement of malnutrition by the MNA scale compared to other assessment measures such as BMI and albumin. This deserves additional studies and we correct this part by specifying this.

Reviewer 2 Report
The manuscript entitled “Evaluation of nutritional status in an acute geriatric unit: prospective study and analysis with frailty syndrome” presents interesting issue, however some corrections are needed.
– There are no line numbers, I cannot write what lines have errors.
– All abbreviations when used for the first time must be explained
– “from November 24, 2014 to May 24, 2015.” – the data is a bit old – what was the reason?
– More information is needed about the validity and reliability of each scale. Additionally, any limitations in reliability and validity need to be addressed in the discussion.
– Tables - A decimal separator should be dot (instead of coma)
– Table 1 – please add the units
– Please emphasize the novelty of the research, because it is missing
– manuscript requires some editorial corrections in formatting
Author Response
RESPONSE TO REVIEWER 2 :
The manuscript entitled “Evaluation of nutritional status in an acute geriatric unit: prospective study and analysis with frailty syndrome” presents interesting issue, however some corrections are needed.
– All abbreviations when used for the first time must be explained
RESPONSE: Correction is done
– “from November 24, 2014 to May 24, 2015.” – the data is a bit old – what was the reason?
RESPONSE: We did this study at that time. We recognize that the study goes back, but the data and their analyzes show both interesting results and allow us to pinpoint the need for a joint evaluation of malnutrition and frailty syndrome in the elderly.
– More information is needed about the validity and reliability of each scale. Additionally, any limitations in reliability and validity need to be addressed in the discussion.
RESPONSE: The scales used in screening for frailty syndrome are all validated, in particular by expert committees such as the Haute Autorité de Santé in France. The FRIED scale remains the reference on the phenotypic side and the Rockwood scale is the reference on the multidimensional side, and the SEGA scale remains a French-speaking multidimensional scale validated in elderly subjects.
The frailty syndrome screening scales have limitations in their use. Indeed, they can be time-consuming like the FRIED scale and the SEGA scale, all the more so as for the FRIED scale which requires the use of a dynamometer. Regarding the CFS or Rockwood scale, it requires in-depth geriatric knowledge and thus requires prior training. It is thus a dependent evaluator.
– Tables - A decimal separator should be dot (instead of coma)
RESPONSE: Correction is done
– Table 1 – please add the units
RESPONSE: We add the units
– Please emphasize the novelty of the research, because it is missing
RESPONSE: It represents a novelty in the sense that several geriatric scales (malnutrition, frailty syndrome, autonomy/dependence, thymia, cognitive assessment and biological markers) have been used together in hospitalized elderly subjects, which was not achieved with this extent in previous scientific work.
